# Cu₂O/CuS/ZnS Nanocomposite Boosts Blue LED-Light-Driven Photocatalytic Hydrogen Evolution

Yu-Cheng Chang *, Yung-Chang Chiao and Ya-Xiu Fun

Department of Materials Science and Engineering, Feng Chia University, Taichung 407, Taiwan
* Correspondence: yuchchang@fcu.edu.tw

**Abstract:** In the present work, we described the synthesis and characterization of the ternary Cu₂O/CuS/ZnS nanocomposite using a facile two-step wet chemical method for blue LED-light-induced photocatalytic hydrogen production. The concentrations of the ZnS precursor and reaction time were essential in controlling the photocatalytic hydrogen production efficiency of the Cu₂O/CuS/ZnS nanocomposite under blue LED light irradiation. The optimized Cu₂O/CuS/ZnS nanocomposite exhibited a maximum photocatalytic hydrogen evolution rate of 1109 $\mu mol h^{-1} g^{-1}$, which was remarkably higher than Cu₂O nanostructures. Through the cycle stability it can be observed that the hydrogen production rate of the Cu₂O/CuS/ZnS nanocomposite decreased after 4 cycles (1 cycle = 3 h), but it remained at 82.2% of the initial performance under blue LED light irradiation. These reasons are mainly attributed to the introduction of CuS and ZnS to construct a rationally coupled reaction system, which enables the synergistic utilization of photogenerated carriers and the increased absorption of visible light for boosting blue LED-light-driven photocatalytic hydrogen evolution.

**Keywords:** Cu₂O/CuS/ZnS nanocomposite; wet chemical; Cu₂O nanostructures; photocatalytic hydrogen production; blue LED light; cycle stability





## 1. Introduction

The widespread use of non-renewable energy sources such as coal and oil harms the Earth's environment and causes energy shortages [1,2]. Therefore, people have begun looking for alternative energy sources that are both environmentally friendly and clean, reducing the demand for non-renewable energy sources [3,4]. In today's renewable energy sources, hydrogen is a colorless, odorless, non-toxic, odorless, and combustible gas whose final product is environmentally friendly water [5–7]. Sunlight is the most inexhaustible energy source on earth [8,9]. If sunlight can be used for photocatalytic water splitting to generate hydrogen, it is the best choice for alternative energy [10,11].

When a semiconductor material is excited by a light source larger than its energy gap, electrons can be excited from the valence band (VB) to the conduction band (CB), forming electron–hole pairs [12]. If there is no other transfer method in the process, it is easy to produce reorganization [13,14]. After the photocatalytic decomposition of water to generate hydrogen, the electron–hole pairs were separated and migrated to the photocatalyst surface. Holes oxidize H₂O to H⁺ and O₂, while electrons reduce H⁺ to H₂. When the valence band edge potential is more positive (>+1.23 eV) than the water oxidation potential, and the conduction band edge is more negative (>−0.01) than the water reduction potential, it may be more favorable for photocatalytic water splitting to produce hydrogen [15,16]. However, it is still limited by the problem of electron–hole pair recombination. This result shows that combining different semiconductor materials can improve photocatalytic hydrogen production efficiency [17–19].

Cuprous oxide (Cu₂O) is a p-type semiconductor with an energy gap of 2.17 eV, which has been widely studied and applied due to its low price, non-toxicity, abundant crustal

content, electrocatalytic, and photocatalytic activities [20,21]. Recently, $Cu_2O$ has been widely studied in many applications, such as gas sensors [22,23], supercapacitors [24,25], lithium-ion batteries [26,27], electrocatalysts [28,29], and photocatalysts to reduce carbon dioxide [30]. However, the poor charge separation ability of pure CuO limits its photocatalytic applications [31]. Therefore, coupling $Cu_2O$ with other semiconductor materials is a feasible strategy to improve the separation rate of photogenerated electron–hole pairs [32,33]. So far, different kinds of CuS–semiconductor composites have been synthesized and have significantly improved the photocatalytic efficiency of single materials, such as $Cu_2O/TiO_2$ [31,34], $Cu_2O/CuO$ [35], $Cu_2O/ZnO$ [33], and $Cu_2O/g-C_3N_4$ [32]. Recently, ternary heterostructures composed of three semiconductors have gained much attention due to their ability to transfer and separate photogenerated electron–hole pairs more efficiently, expand the photoresponse range, and significantly improve photocatalytic performance, such as $Cu_2O-Co_3O_4/CN$ [36], $MoS_2-CdS-Cu_2O$ [37], $Cu(OH)_2/Cu_2O/C_3N_4$ [38], and $ZnO/Cu_2O-CuO$ [39]. However, to our knowledge, no literature has ever reported that the synthesis of the $Cu_2O/CuS/ZnS$ nanocomposite has been applied in photocatalytic water splitting.

Herein, a ternary $Cu_2O/CuS/ZnS$ nanocomposite can be synthesized by a facile two-step wet chemical process to validate as a highly efficient water-splitting catalyst under blue LED light irradiation. The improved photocatalytic activities and photostability of the $Cu_2O/CuS/ZnS$ nanocomposite can be attributed to the highly visible light harvesting and efficient separation of the photogenerated electron–hole pairs prompted by the heterojunction structures.

## 2. Results and Discussion

X-ray diffraction (XRD) measurements can confirm the crystalline structures of as-prepared photocatalysts. Figure 1a shows the XRD pattern of $Cu_2O$ nanostructures with sharp and strong peaks indicating the high crystallinity of $Cu_2O$ crystals. The diffraction peaks of $Cu_2O$ nanostructures at $2\theta$ = 29.6°, 36.4°, 42.3°, 61.4°, 73.6°, and 77.4° can be indexed to cubic phase $Cu_2O$ (JCPDS no. 78-2076), corresponding to the (110), (111), (200), (220), (311), and (222) crystal planes, respectively. In addition, there are two weak diffraction peaks at $2\theta$ = 35.6° and 38.8°, which can be indexed to monoclinic phase CuO (JCPDS no. 80-1268), corresponding to the (−111) and (111) planes, respectively. Figure 1b shows the XRD pattern of the $Cu_2O/CuS/ZnS$ nanocomposite with small and broad peaks. After the reaction of ZnS precursors, the main as-grown $Cu_2O$ nanostructures were transformed into the hexagonal CuS crystal phase. Only one diffraction peak of $Cu_2O$ nanostructures at $2\theta$ = 36.4° corresponds to the (111) plane of cubic phase $Cu_2O$ (JCPDS no. 78-2076) that was observed. This phenomenon is due to the Kirkendall effect [40]. The formation of the hexagonal CuS crystal phase corresponding to the (102), (103), (107), (108), (200), (202), and (109) planes, respectively, was verified from the diffraction peaks at 29.6°, 32.1°, 48.2°, 53.3°, 56.5°, 57.8, and 58.6° (JCPDS no. 75-2235). The other two diffraction peaks at $2\theta$ = 28.5° and 33.1° can be indexed to cubic phase ZnS (JCPDS no. 77-2100), corresponding to the (111) and (200) crystal planes, respectively. No additional diffraction peaks are present, suggesting the $Cu_2O/CuS/ZnS$ nanocomposite is composed of $Cu_2O$, CuS, and ZnS.

Field-emission scanning electron microscopy (FE-SEM) and field-emission transmission electron microscopy (FE-TEM) can be used to explore the morphological and microstructural features of the as-prepared photocatalysts. Figure 2a shows the top-view FE-SEM image of $Cu_2O$ nanostructures grown via the wet chemical method under the low reaction temperature of 50 °C for 1 h. $Cu_2O$ nanostructures revealed a more scattered size and shape. The FE-TEM image (Figure 2b) further confirms the microstructures of $Cu_2O$ nanostructures showing apparent aggregation, which is consistent with the FE-SEM image. The selected area electron diffraction (SAED) pattern (Figure 2c) of $Cu_2O$ nanostructures exhibited polycrystalline diffraction rings. The concentric rings (from inside to outside) can, respectively, be indexed to the cubic phase $Cu_2O$ (JCPDS no. 78-2076) and monoclinic phase CuO (JCPDS no. 80-1268) [41]. This result is also consistent with the above XRD

result. In the high-resolution TEM (HRTEM) image (Figure 2d), the lattice fringes of 0.301, 0.246, and 0.252 nm match with the (110) and (111) planes of cubic phase $Cu_2O$ (JCPDS no. 78-2076) and the (−111) plane of monoclinic phase CuO (JCPDS no. 80-1268), respectively, which agrees with the above result of Figure 2c.

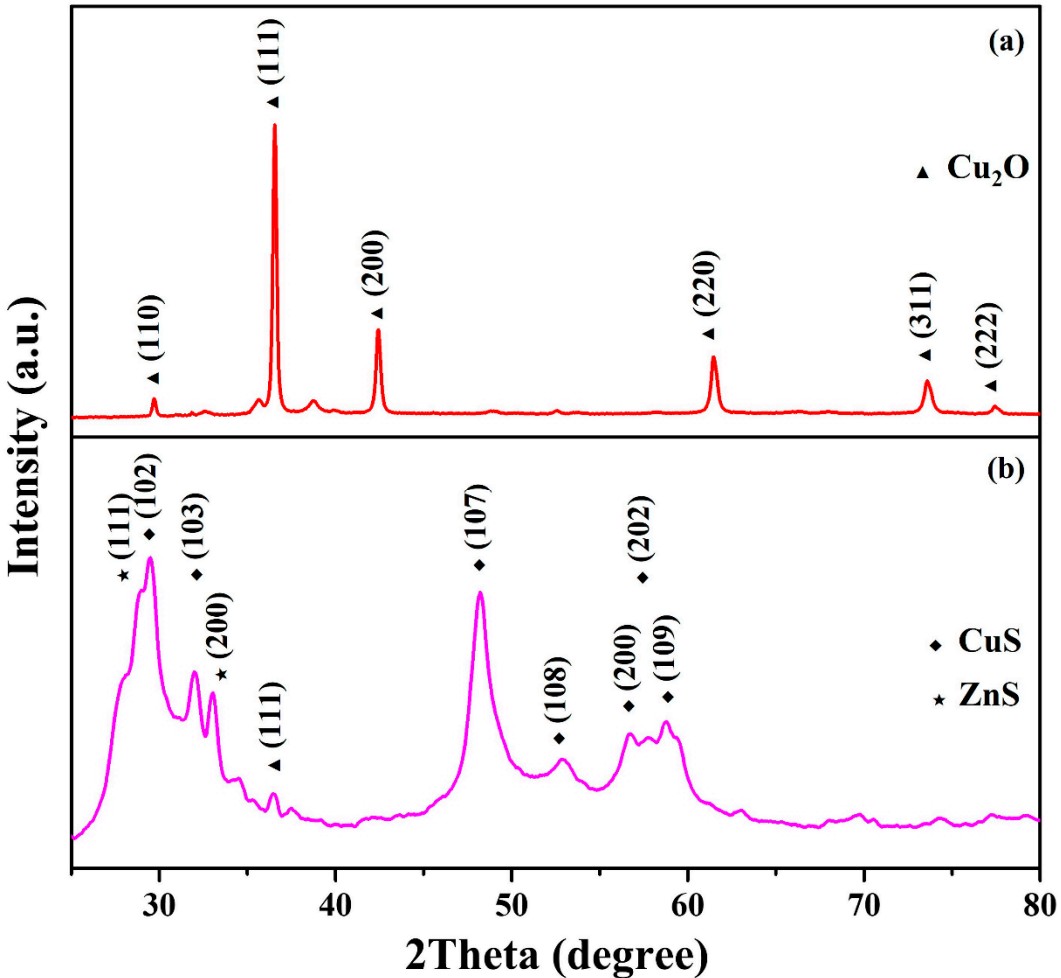

**Figure 1.** The XRD spectra of (**a**) $Cu_2O$ nanostructures and (**b**) the $Cu_2O/CuS/ZnS$ nanocomposite grown at the 10 mM ZnS precursor and 20 min reaction time.

Figure 3a shows the top-view FE-SEM image of $Cu_2O/CuS/ZnS$ nanocomposite with 0.02 g $Cu_2O$ nanostructures and 10 mM ZnS precursor grown via the wet chemical method at 100 °C for 1 h. Compared with $Cu_2O$ nanostructures, $Cu_2O/CuS/ZnS$ nanocomposite tends to be smaller and more uniform in size. Furthermore, the FE-TEM image (Figure 3b) further confirms the microstructures of the $Cu_2O/CuS/ZnS$ nanocomposite, showing an evident aggregation phenomenon between particles, which is consistent with the FE-SEM image. The SAED pattern (Figure 3b inset) of the $Cu_2O/CuS/ZnS$ nanocomposite also revealed polycrystalline diffraction rings. The concentric rings (from inside to outside) can, respectively, be indexed to the cubic phase ZnS (JCPDS no. 77-2100), hexagonal phase CuS (JCPDS no. 75-2235), and cubic phase $Cu_2O$ (JCPDS no. 78-2076). Three lattice spacing measurements of 0.319, 0.313, and 0.246 nm can be detected from the HRTEM image in Figure 3c, which corresponds to the (101), (111), and (111) diffraction planes of hexagonal phase CuS, cubic phase ZnS, and cubic phase $Cu_2O$, respectively. The EDS elemental mapping images (Figure 3d) further confirmed that Cu, O, Zn, and S were present and evenly distributed throughout the $Cu_2O/CuS/ZnS$ nanocomposite.

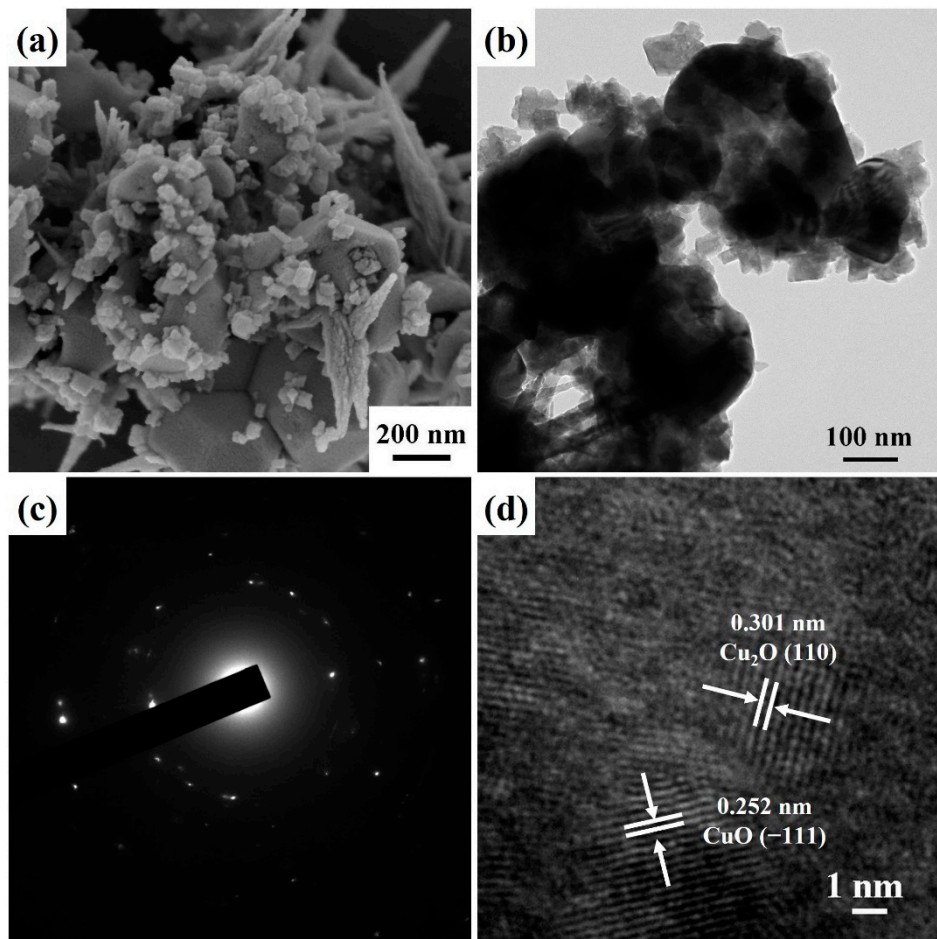

**Figure 2.** (**a**) FE-SEM, (**b**) FE-TEM, (**c**) SAED pattern, and (**d**) HRTEM images of $Cu_2O$ nanostructures.

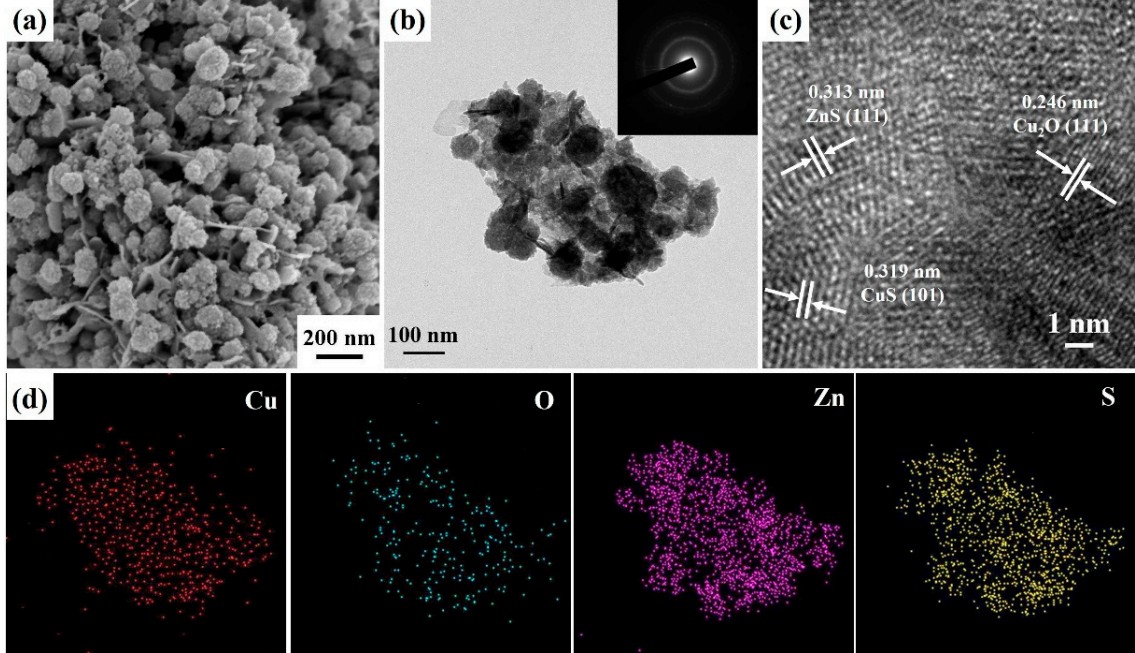

**Figure 3.** (**a**) FE-SEM, (**b**) FE-TEM and SAED pattern (inset), (**c**) HRTEM, and (**d**) EDS mapping images of the $Cu_2O/CuS/ZnS$ nanocomposite grown at the 10 mM ZnS precursor and 20 min reaction time.

X-ray photoelectron spectroscopy (XPS) analysis can be used to verify the surface composition and bonding of the $Cu_2O/CuS/ZnS$ nanocomposite, as shown in Figure 4. The survey XPS spectrum (Figure 4a) demonstrates that $Cu_2O/CuS/ZnS$ nanocomposite includes Cu, O, Zn, and S elements. The presence of C 1s can be attributed to the pump oil in the vacuum system of the XPS device or to the organic layer coated on the $Cu_2O/CuS/ZnS$ nanocomposite [14]. The high-resolution $Cu2p_{3/2}$ spectrum (Figure 4b) shows that two peaks at 931.7 eV and 932.6 eV correspond to the different oxidation states of $Cu^+$ and $Cu^{2+}$ ions for $Cu_2O$ and CuS, respectively. In addition, the high-resolution $Cu2p_{1/2}$ spectrum (Figure 4c) also reveals that two peaks at 951.7 eV and 953.2 eV correspond to the different oxidation states of $Cu^+$ and $Cu^{2+}$ ions for $Cu_2O$ and CuS, respectively [42,43]. According to the O 1s spectrum (Figure 4d), there are three peaks at 530.4 eV, 531.5 eV, and 532.4 eV, which are attributed to Cu−O−Cu (lattice O, $O_L$), oxygen vacancies or defect ($O_V$), and chemisorbed or dissociated ($O_C$), respectively [17,44]. The peaks of Figure 4e at 1021.8 and 1044.9 eV belong to Zn $2p_{3/2}$ and Zn $2p_{1/2}$, respectively, suggesting the presence of ZnS [45,46]. According to the S 2p spectrum (Figure 4f), the peaks at 161.5 and 162.6 eV can be assigned to S $2p_{3/2}$ and S $2p_{1/2}$, which indicates that In exists as $S^{2-}$ [47]. These results confirm that the $Cu_2O/CuS/ZnS$ nanocomposite was successfully constructed.

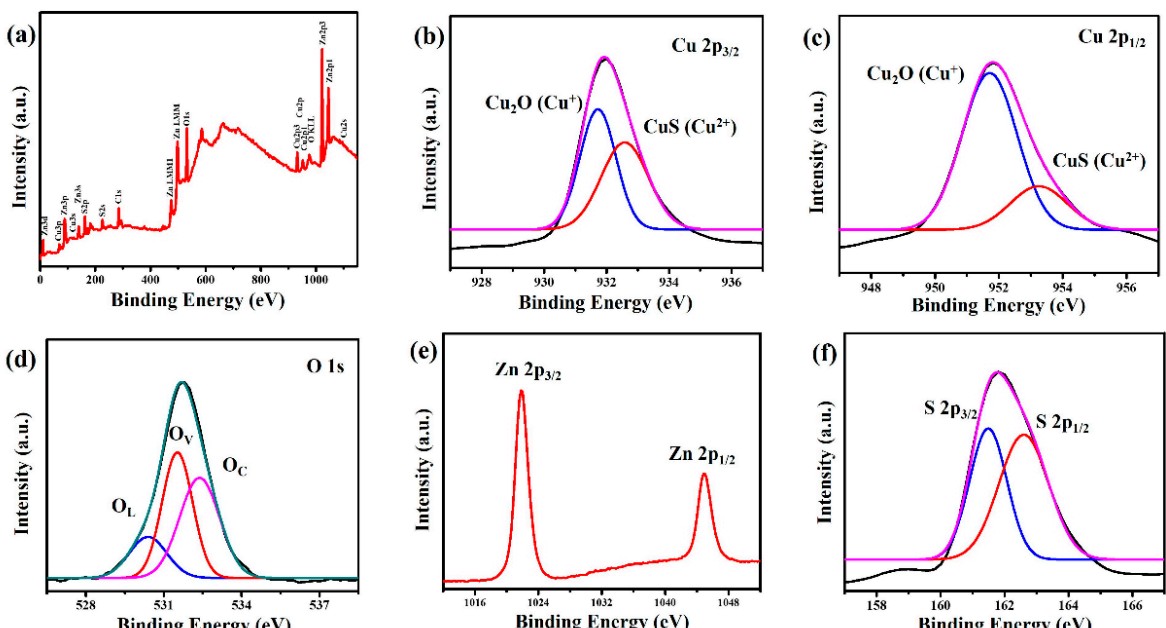

**Figure 4.** The XPS spectra of the $Cu_2O/CuS/ZnS$ nanocomposite: (**a**) survey spectrum, (**b**) Cu $2p_{3/2}$, (**c**) Cu $2p_{1/2}$, (**d**) O 1s, (**e**) Zn 2p, and (**f**) S 2p.

In order to evaluate the photocatalytic performance of the $Cu_2O/CuS/ZnS$ nanocomposite, blue LED-light-driven hydrogen production via water splitting was accomplished using sodium sulfide as a sacrificial reagent. Figure 5a shows the comparable photocatalytic hydrogen production activities of the $Cu_2O/CuS/ZnS$ nanocomposite grown at different concentrations of ZnS precursors (such as 2.5, 5, 10, and 20 mM). The as-synthesized photocatalysts' average hydrogen evolution rates (HER) are 0 ($Cu_2O$ nanostructures), 6.433 (2.5 mM ZnS precursor), 120.9 (5 mM ZnS precursor), 601.2 (10 mM ZnS precursor), and 388.6 (20 mM ZnS precursor) $\mu mol h^{-1} g^{-1}$. No hydrogen can be detected in the $Cu_2O$ nanostructures. $Cu_2O$ nanostructures exhibited inferior hydrogen production activities, attributed to the fast recombination of photogenerated electron–hole pairs [31,33]. The HER of $Cu_2O/CuS/ZnS$ nanocomposites gradually increased with ZnS precursor concentration. The HER of the $Cu_2O/CuS/ZnS$ nanocomposite decreased significantly with a ZnS precursor concentration greater than 10 mM. This phenomenon is also consistent with the previous literature [48]. Figure 5b compares the photocatalytic hydrogen production

efficiency of the $Cu_2O/CuS/ZnS$ nanocomposite grown at different reaction times. The average HERs of the $Cu_2O/CuS/ZnS$ nanocomposite are 473.0 (10 min), 1109 (20 min), 870.8 (30 min), and 716.5 (40 min) $\mu molh^{-1}g^{-1}$. The HER of the $Cu_2O/CuS/ZnS$ nanocomposite gradually increased with the reaction time from 10 to 20 min. However, the further increase resulted in a decreased HER of the $Cu_2O/CuS/ZnS$ nanocomposite. This result may be attributed to the fact that too long a reaction time may lead to the complete reaction of $Cu_2O$ to form CuS, which in turn reduces the efficiency of light absorption and electron and hole transfer, thereby inhibiting the efficiency of photocatalytic hydrogen production.

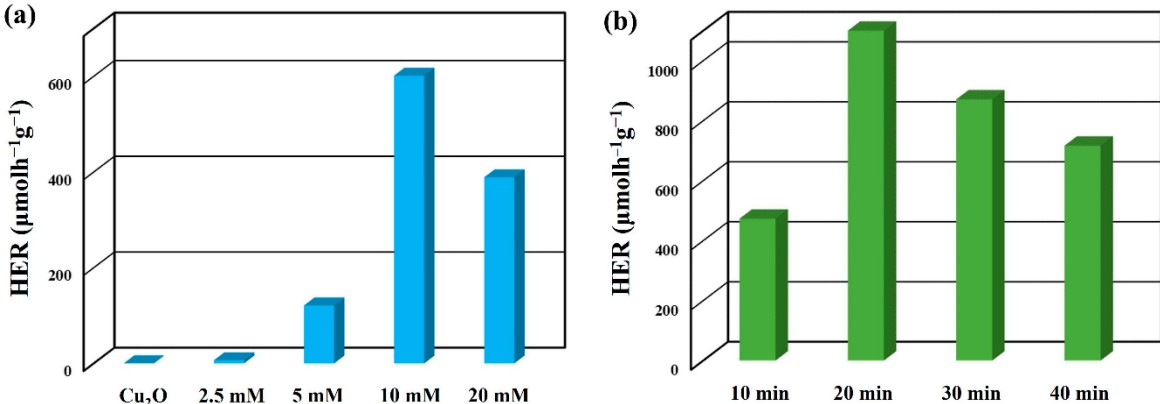

**Figure 5.** The average HER of the $Cu_2O/CuS/ZnS$ nanocomposite grown at different (**a**) ZnS precursor concentrations and (**b**) reaction times under blue LED light irradiation.

The optical absorption properties of the $Cu_2O$ nanostructures and $Cu_2O/CuS/ZnS$ nanocomposite were evaluated by UV–Vis DRS spectroscopy, as shown in Figure 6a. Compared with $Cu_2O$ nanostructures, the $Cu_2O/CuS/ZnS$ nanocomposite exhibited stronger absorption bands in the visible region. This result reveals that the decoration of CuS and ZnS can significantly increase the visible light absorption (475–800 nm) of the $Cu_2O/CuS/ZnS$ nanocomposite, thereby improving its photocatalytic hydrogen production. PL emission peaks on semiconductor materials mainly originate from photoinduced electron/hole pair recombination [49]. $Cu_2O$ nanostructures exhibit a stronger green emission peak at 532 nm (2.33 eV), which is attributed to the electron–hole pair recombination of the near-bandgap emission (NBE) of $Cu_2O$ [50,51]. When $Cu_2O$ is combined with CuS and ZnS, the photoinduced carriers can migrate between the composite materials, thereby inhibiting the photoinduced electron–hole pair recombination. This result is beneficial in enhancing the photocatalytic hydrogen production efficiency of the $Cu_2O/CuS/ZnS$ nanocomposite.

Figure 6c shows the possible photocatalytic hydrogen production mechanism of the $Cu_2O/CuS/ZnS$ nanocomposite for photocatalytic hydrogen production by the above analysis results. Ion exchange resin coats the materials $Cu_2O$, CuS, and ZnS in indium tin oxide (ITO) glass and then measures the flat band potential by cyclic voltammogram [52,53]. The VB and CB of $Cu_2O$, CuS, and ZnS are consistent with the previous reports [54,55]. The $Cu_2O$, CuS, and ZnS CB positions are $-1.4$ eV, $-0.5$ eV, and $-0.99$ eV, respectively. The $Cu_2O$, CuS, and ZnS VB positions are 0.93 eV, 1.91 eV, and 2.52 eV, respectively. Intrinsic defects can generate new electric state bands at the bottom of the CB of ZnS, combined with narrowing the band gap to enhance visible light absorption. The photogenerated electrons of $Cu_2O$, CuS, and ZnS can be excited from their VB to CB under visible light irradiation. The photogenerated electrons in the CB of ZnS and $Cu_2O$ can be transferred to the CB of CuS. CuS can act as an electron sink to capture and reduce hydrogen ions to hydrogen. Meanwhile, the photogenerated holes in the VB of ZnS can be transferred to the CuS or $Cu_2O$ to oxidize water to oxygen or hydrogen ions. Therefore, this photocatalytic process can improve the separation of photogenerated charge carriers and facilitate their photocatalytic hydrogen production.

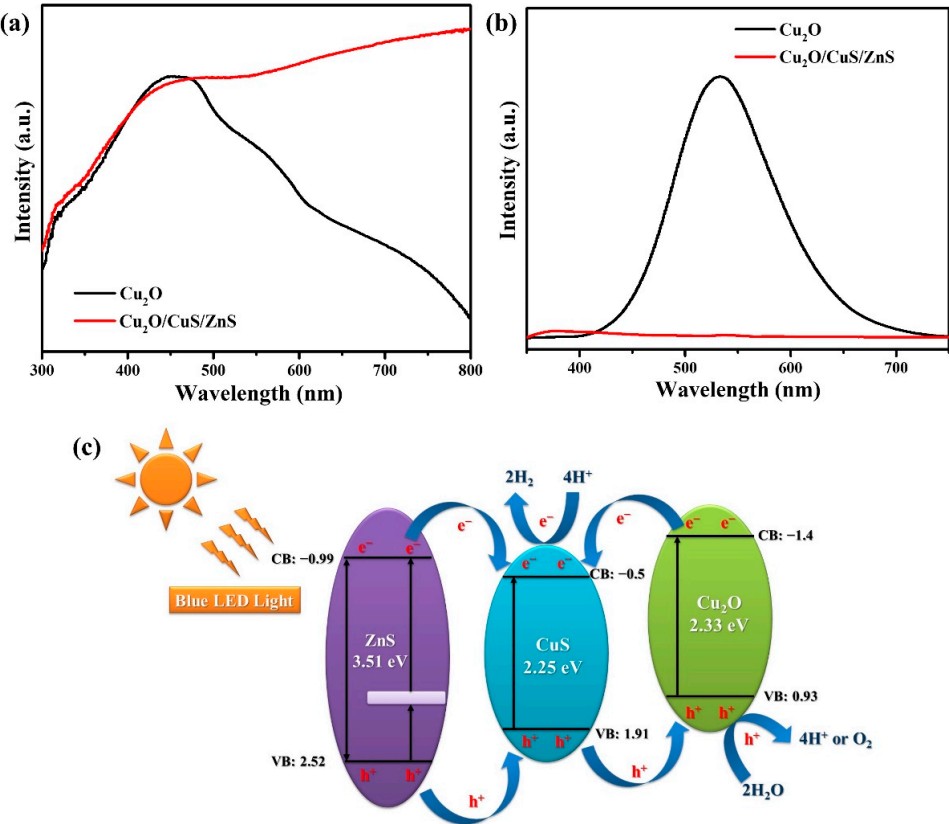

**Figure 6.** The (**a**) UV–Vis absorption and (**b**) PL spectra of $Cu_2O$ nanostructures and the $Cu_2O/CuS/ZnS$ nanocomposite. (**c**) Schematic diagram of the electron transfer mechanism of the $Cu_2O/CuS/ZnS$ nanocomposite under blue LED light irradiation.

Sacrificial reagents are often used for the photocatalytic splitting of water to generate hydrogen to enhance the performance of oxidation reactions in aqueous media because pure water oxidation is usually inefficient [56]. Figure 7a displays the effects of four sacrificial agents (such as folic acid, methanol, sodium sulfate, and sodium sulfide) on the photocatalytic efficiency of the optimized $Cu_2O/CuS/ZnS$ nanocomposite (10 mM ZnS precursor and 20 min reaction time) without adjusted pH value. All sacrificial agents were formulated at the same 0.1 M concentration. The order of average HER with the $Cu_2O/CuS/ZnS$ nanocomposite is sodium sulfide (622.7 $\mu molh^{-1}g^{-1}$) > folic acid (4.753 $\mu molh^{-1}g^{-1}$) > sodium sulfate (0 $\mu molh^{-1}g^{-1}$) = methanol (0 $\mu molh^{-1}g^{-1}$). There are two main reasons for using sodium sulfide as a sacrificial reagent for the best hydrogen production efficiency. First, sulfide ions ($S^{2-}$, dissociated sodium sulfide) can be adsorbed on the photocatalyst and react with photogenerated holes, thereby inhibiting the recombination of electron and hole pairs. Second, sulfide ions can reduce the photocorrosion rate of metal sulfide semiconductors by combining with metal ions and inhibiting sulfide defects, thereby improving the stability of photocatalytic hydrogen production [57,58].

The optimum pH value of photocatalytic hydrogen production mainly depends on the properties of the sacrificial agent and the adsorption on the surface of the photocatalyst [59]. The different pH values can be adjusted from their initial value by adding dropwise 1M HCl. Figure 7b displays the effects of pH values on the photocatalytic efficiency of the optimized $Cu_2O/CuS/ZnS$ nanocomposite (10 mM ZnS precursor and 20 min reaction time). The average HERs of the $Cu_2O/CuS/ZnS$ nanocomposite are 0 (pH = 3), 336.2 (pH = 6), 473.5 (pH = 9), 1109 (pH = 12), and 622.8 $\mu molh^{-1}g^{-1}$ (pH = 12.8, without adjusted), respectively. From pH 3 to 12, the photocatalytic hydrogen production efficiency revealed significant improvement with the gradual increase in pH value. This phenomenon can be mainly attributed to the dissociation of $HS^-$ and $S^{2-}$, which increases gradually

with increased pH value [60]. When the concentration of hydroxide ions is too high, many photogenerated hydrogen ions can further react with hydroxide ions to form water, thereby reducing the photocatalytic hydrogen production efficiency [61]. This result indicates that the Cu$_2$O/CuS/ZnS nanocomposite exhibits the best photocatalytic hydrogen production efficiency at pH = 12.

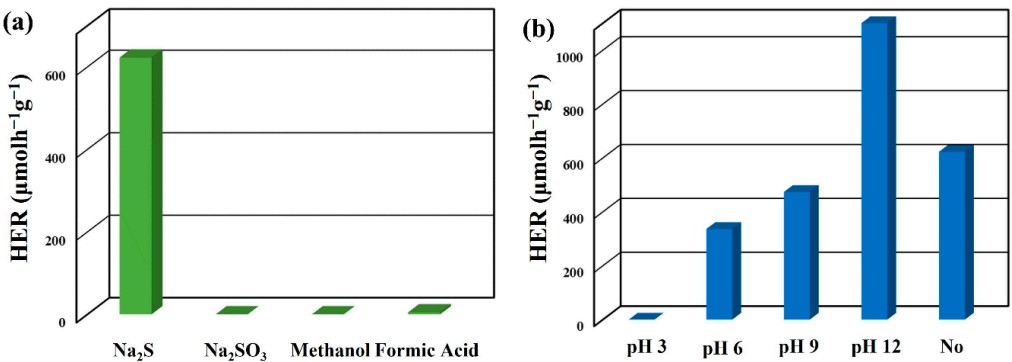

**Figure 7.** The average HER of optimized Cu$_2$O/CuS/ZnS nanocomposite at (**a**) sacrificial reagents and (**b**) different pH values.

The practicality of photocatalysts mainly depends on their stability and reusability [62]. Therefore, it was used consecutively to evaluate the photocatalyst's stability to ensure its recyclability in photocatalytic experiments. In addition, the light source is the main factor due to the overall photocatalytic efficiency depending on the irradiation intensity [63]. Herein, the reusability of the optimized Cu$_2$O/CuS/ZnS nanocomposite is also evaluated by performing the water-splitting reaction for 3 h with four cycles under blue LED light (Figure 8a) and white LED light (Figure 8b) irradiation. As a result, the average HERs of optimized the Cu$_2$O/CuS/ZnS nanocomposite are 1025, 1029, 911.2, and 851.6 μmolh$^{-1}$g$^{-1}$, respectively, under blue LED light irradiation. On the other hand, the average HERs of the optimized Cu$_2$O/CuS/ZnS nanocomposite are 647.8, 493.8, 483.9, and 407.4 μmolh$^{-1}$g$^{-1}$, respectively, under white LED light irradiation. After the reusability tests, the photocatalytic hydrogen production efficiency of the Cu$_2$O/CuS/ZnS nanocomposite revealed a slight decrease ($\sim$17.8% and $\sim$37.1%) under the blue LED light and white LED light irradiation, respectively. This result shows that the Cu$_2$O/CuS/ZnS nanocomposite exhibited the best photocatalytic hydrogen generation efficiency and recycle stability under blue LED light irradiation. The possible reason is that the energy of the white LED light is dispersed in two prominent bands, but blue light only has a single band. Therefore, this band's energy can be concentrated, increasing the photocatalytic hydrogen production efficiency.

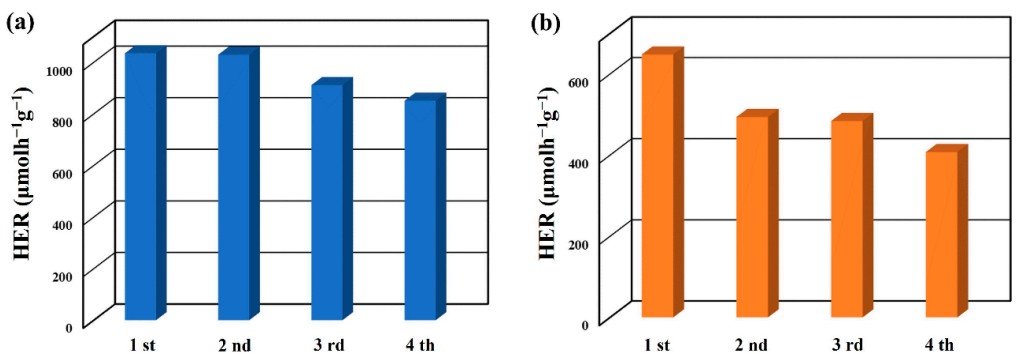

**Figure 8.** The cycling stability of the optimized Cu$_2$O/CuS/ZnS nanocomposite under (**a**) blue LED light and (**b**) white LED light irradiation.

## 3. Material and Methods

### 3.1. Preparation of $Cu_2O$ Nanostructures

$Cu_2O$ nanostructures were synthesized by a simple wet chemical method. In a typical procedure, 2.5 mL 0.1 M $CuCl_2 \cdot 2H_2O$, 0.9 mL 0.1 M NaOH, 12 mL 0.1 M $NH_2OH \cdot HCl$, and 0.435 g SDS were dispersed in 34.6 mL deionized water. The mixture was heated at 50 °C under vigorous stirring for 1 h. After cooling, the as-prepared products were collected, washed with deionized water, and dried at 70 °C for 2 h.

### 3.2. Preparation of the $Cu_2O/CuS/ZnS$ Nanocomposite

The $Cu_2O/CuS/ZnS$ nanocomposite was synthesized using a simple wet chemical method. In a typical procedure, different concentrations of ZnS precursor (equimolar zinc nitrate hexahydrate and thioacetamide) were dissolved in 100 mL of deionized water. The 0.02 g $Cu_2O$ nanostructures were dispersed in a 100 mL reaction solution with different concentrations of ZnS precursor. The mixture was heated at 100 °C under vigorous stirring for 1 h. After cooling, the as-prepared products were collected, washed with deionized water, and dried at 70 °C for 2 h.

### 3.3. Characterization

The phase structure, morphology, microstructure, and optical properties of as-synthesized photocatalysts were, respectively, measured by X-ray powder diffraction (Bruker D2 phaser system, USA), field-emission scanning electron microscopy (FESEM, Hitachi S-4800, Japan), field-emission transmission electron microscopy (FETEM, JEOL-2100F, Japan), X-ray photoelectron spectroscopy (XPS, ULVAC-PHI PHI 5000 Versaprobe II system, Japan), photoluminescence (PL, 325 nm He-Cd laser, Taiwan), and UV–Vis DRS spectroscopy (U-2900, Hitachi, Japan).

### 3.4. Photocatalytic Activity Measurement

A photocatalytic reactor (PCX50B Discover, Perfect Light, China) with blue LED light sources (5W, $\lambda_{max}$ = 420 nm) or white LED light (5W, two $\lambda_{max}$ at 450 nm and 550 nm) was used to evaluate photocatalytic hydrogen production. In a typical experiment, as-prepared photocatalysts (20 mg) were dispersed into a mixed solution with 0.1 M sacrificial reagents (such as sodium sulfide, sodium sulfite, methanol, and ethanol) and 50 mL of deionized water ($\geq$18.3 M$\Omega \cdot$cm). After a degassing pretreatment, the experimental device was carried out for 30 min to remove air. Then, the hydrogen amounts were measured using gas chromatography (GC, Shimadzu GC-2014) with a thermal conductivity detector (TCD).

## 4. Conclusions

A facile two-step wet chemical process synthesizes a new noble metal-free heterostructure of $Cu_2O/CuS/ZnS$ nanocomposite for blue LED-light-induced photocatalytic hydrogen production. The optimized $Cu_2O/CuS/ZnS$ nanocomposite could achieve $H_2$ evolution rates of up to 1109 $\mu$molh$^{-1}$g$^{-1}$ at a pH value = 12 and 0.1 M sodium sulfide under blue LED light irradiation. This result can be attributed to the formation of CuS and ZnS on $Cu_2O$ nanostructures, which can effectively absorb visible light emission and promote the separation efficiency of photogenerated electron–hole pairs under blue LED light irradiation. Furthermore, reusability experiments prove that the $Cu_2O/CuS/ZnS$ nanocomposite exhibits excellent stability for a long-term photocatalytic process under blue LED light irradiation.

**Author Contributions:** Funding acquisition, methodology, project administration, resources, software, supervision, validation, writing—original draft, and writing—review and editing, Y.-C.C. (Yu-Cheng Chang); formal analysis, investigation, and data curation, Y.-C.C. (Yung-Chang Chiao) and Y.-X.F. All authors have read and agreed to the published version of the manuscript.

**Funding:** This research was funded by the Ministry of Science and Technology of Taiwan (MOST 109-2221-E-035-041-MY3).

**Data Availability Statement:** Not applicable.

**Acknowledgments:** The authors appreciate the Precision Instrument Support Center of Feng Chia University for providing the fabrication and measurement facilities.

**Conflicts of Interest:** The authors declare no conflict of interest.

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
