# Peer review of "Cu2O/CuS/ZnS Nanocomposite Boosts Blue LED-Light-Driven Photocatalytic Hydrogen Evolution"

_catalysts, doi:10.3390/catal12091035_

Round 1

Reviewer 1 Report

This work reports a route to synthesize a new noble metal-free hetero-structure of Cu2O/CuS/ZnS nanocomposite by a facile two-step wet-chemical process. The optimized Cu2O/CuS/ZnS nanocomposite showed good photocatalytic hydrogen production under blue LED. Although some results seem attractive, however, I cannot recommend it for publication in its present form. Detailed comments are as following.

1.In the XRD characterization, there isn’t any signal of Cu2O diffraction peaks in the XRD pattern of Cu2O/CuS/ZnS nanocomposite. And the further EDS results could not give a strong evidence whether the Cu2O are still existing in the nanocomposite or not. It would be better if the authors could further prove the composition of the nanocomposite.

2.In XPS analysis, there is much difference in the fitted line from the raw data, with the width of split peak too big. It would be better if the authors could further optimize and decorate the profile.

3.In the PHE(Photocatalyst hydrogen production) experiments, the average HER of Cu2O/CuS/ZnS nanocomposite grown at different reaction times shows that the photocatalyst will increase at the first 20 min and decrease dramatically after 30 min. It means the photocatalyst doesn’t show a good stability. And the reaction time is too low, so it is better to take a longer time as an observation interval.  

4.In the mechanism explanation part, the authors give an energy band structure of the three materials. If possible, the authors should add some experiments to show how the energy band and conductive band were calculated and concluded.

5.In the pH experiment part, details such as how the pH value was adjusted and other conditions were controlled should be added to make the experiment more complete and convincing.

Author Response

This work reports a route to synthesize a new noble metal-free hetero-structure of Cu2O/CuS/ZnS nanocomposite by a facile two-step wet-chemical process. The optimized Cu2O/CuS/ZnS nanocomposite showed good photocatalytic hydrogen production under blue LED. Although some results seem attractive, however, I cannot recommend it for publication in its present form. Detailed comments are as following.

Response: Thanks for the pertinent and positive comments.

  1. In the XRD characterization, there isn’t any signal of Cu2O diffraction peaks in the XRD pattern of Cu2O/CuS/ZnS nanocomposite. And the further EDS results could not give a strong evidence whether the Cu2O are still existing in the nanocomposite or not. It would be better if the authors could further prove the composition of the nanocomposite.

Response: Thanks for your reminder. XRD spectrum (Fig. 1b only one Cu2O plane (111)) XPS spectra (Fig. 4b and c) and HRTEM image (Fig. 3c) can use to prove that the composite contains Cu2O.

  1. In XPS analysis, there is much difference in the fitted line from the raw data, with the width of split peak too big. It would be better if the authors could further optimize and decorate the profile.

Response: Thanks for your reminder. We have re-optimized and decorated the profile of O 1s in the new Fig. 4d.

  1. In the PHE(Photocatalyst hydrogen production) experiments, the average HER of Cu2O/CuS/ZnS nanocomposite grown at different reaction times shows that the photocatalyst will increase at the first 20 min and decrease dramatically after 30 min. It means the photocatalyst doesn’t show a good stability. And the reaction time is too low, so it is better to take a longer time as an observation interval.

Response: Thanks for your reminder. This suggestion will incorporate future research on long-reaction times in related studies.

  1. In the mechanism explanation part, the authors give an energy band structure of the three materials. If possible, the authors should add some experiments to show how the energy band and conductive band were calculated and concluded.

Response: Thanks for your reminder. Ion exchange resin coats the materials Cu2O, CuS, and ZnS on indium tin oxide (ITO) glass and then measure the flat band potential by cyclic voltammogram [1,2]. The VB and CB of Cu2O, CuS, and ZnS are consistent with the previous reports [3,4].

  1. In the pH experiment part, details such as how the pH value was adjusted and other conditions were controlled should be added to make the experiment more complete and convincing.

Response: Thanks for your reminder. The different pH values can be adjusted from their initial value by dropwise adding 1M HCl.

  1. Ge, H.; Tian, H.; Zhou, Y.; Wu, S.; Liu, D.; Fu, X.; Song, X.-M.; Shi, X.; Wang, X.; Li, N. Influence of Surface States on the Evaluation of the Flat Band Potential of TiO2. ACS Appl. Mater. Interfaces 2014, 6, 2401-2406, doi:10.1021/am404743a.
  2. Bhattacharya, C.; Lee, H.C.; Bard, A.J. Rapid Screening by Scanning Electrochemical Microscopy (SECM) of Dopants for Bi2WO6 Improved Photocatalytic Water Oxidation with Zn Doping. J. Phys. Chem. C 2013, 117, 9633-9640, doi:10.1021/jp308629q.
  3. Gu, Y.; Xu, Z.; Guo, L.; Wan, Y. ZnO nanoplate-induced phase transformation synthesis of the composite ZnS/In(OH)3/In2S3 with enhanced visible-light photodegradation activity of pollutants. CrystEngComm 2014, 16, 10997-11006, doi:10.1039/C4CE01922A.
  4. Huang, J.-Y.; Hsieh, P.-L.; Naresh, G.; Tsai, H.-Y.; Huang, M.H. Photocatalytic Activity Suppression of CdS Nanoparticle-Decorated Cu2O Octahedra and Rhombic Dodecahedra. J. Phys. Chem. C 2018, 122, 12944-12950, doi:10.1021/acs.jpcc.8b03609.

Reviewer 2 Report

The titled work “Cu2O/CuS/ZnS nanocomposite boosts blue LED-light-driven photocatalytic hydrogen evolution” was reviewed. The work is appropriated to publish in catalysts; however, there are some issues to attend before the publication.

Results of HER by each (by separated) of photocatalysts should be considered for comparing the experimental results.

What does “ascorbic to cubic” phase mean?

In figure 1b the signals of Cu2O are not observed totally. Authors indicated that Cu2O was transformed to CuS because only one plane (111) was identified. Authors need to explain this reaction under the thermodynamic and kinetic perspectives. Another possibility is the preferential orientation in the XRD measuring due to inadequate position of the sample in the equipment.

In general, in TEM results explanation authors need to refer other works for supporting them.

Line 138: change ESD by EDS.

What was the reason to use Na2SO3, Methanol and formic acid as sacrificial agent?

By extrapolation, in the cycle number 10 the HER will be 50% and zero, respectively. What is the opinion of the authors?

Author Response

The titled work “Cu2O/CuS/ZnS nanocomposite boosts blue LED-light-driven photocatalytic hydrogen evolution” was reviewed. The work is appropriated to publish in catalysts; however, there are some issues to attend before the publication.

Response: Thanks for the pertinent and positive comments.

Results of HER by each (by separated) of photocatalysts should be considered for comparing the experimental results.

Response: Thanks for your reminder. We have also tried to measure the photocatalytic hydrogen production efficiency of pure CuS and ZnS nanopowder. This result revealed the same with Cu2O nanostructures (0 µmolh−1g−1) under the same photocatalytic reaction process.

What does “ascorbic to cubic” phase mean?

Response: Thanks for your reminder. This oversight has been amended in the revised manuscript.

In figure 1b the signals of Cu2O are not observed totally. Authors indicated that Cu2O was transformed to CuS because only one plane (111) was identified. Authors need to explain this reaction under the thermodynamic and kinetic perspectives. Another possibility is the preferential orientation in the XRD measuring due to inadequate position of the sample in the equipment.

Response: Thanks for your reminder. XPS spectra (Fig. 4b and c) and HRTEM image (Fig. 3c) can use to prove that the composite contains Cu2O. Furthermore, a simple Kirkendall-based process using the prepared CuO nanostructures as sacrificial templates has been successfully used to prepare Cu2O/CuS/ZnS nanocomposite [1].

In general, in TEM results explanation authors need to refer other works for supporting them.

Response: Thanks for your reminder. We have updated some references in the revised manuscript.

Line 138: change ESD by EDS.

Response: Thanks for your reminder. This oversight has been amended in the revised manuscript.

What was the reason to use Na2SO3, Methanol and formic acid as sacrificial agent?

Response: Thanks for your reminder. The selection of sacrificial reagents is mainly based on the previous literature on metal oxides and sulfides in hydrogen production, and several of them are selected for research and discussion.

By extrapolation, in the cycle number 10 the HER will be 50% and zero, respectively. What is the opinion of the authors?

Response: Thanks for your reminder. This phenomenon is possible. It is speculated that the reduction of hydrogen production efficiency may be caused by the loss of photocatalysts under multiple centrifugations.

  1. Cai, L.; Sun, Y.; Li, W.; Zhang, W.; Liu, X.; Ding, D.; Xu, N. CuS hierarchical hollow microcubes with improved visible-light photocatalytic performance. RSC Advances 2015, 5, 98136-98143, doi:10.1039/C5RA18563G.

Reviewer 3 Report

The present study has described the synthesis and characterization of ternary Cu2O/CuS/ZnS nanocomposite using a two-step wet-chemical method for blue LED light-induced photocatalytic hydrogen production. The noble metal-free ternary nanocomposite has shown high photocatalytic activity in hydrogen evolution reaction. The manuscript is suggested for publication after minor modifications.

1. The Cu-containing ternary catalyst shows high activity in photocatalytic hydrogen evolution. In order to demonstrate this, it would be very useful to compare the archived maximum value to that obtained on other noble metal-free ternary catalyst for example the recently obtained nickel phosphide anchored on anatase-hematite heterojunction (Int. Journal of Hydrogen Energy 47, (2022) 23593-23607). You can do it in the Introduction or in the Conclusion.

2. The differentiation between Cu1+ and Cu2+ in XPS without analyzing the Auger transitions is very difficult.

3. The XPS technique is not mentioned in the 3.3 Characterization section.

4. Please indicate the BE reference in XPS (residual C?)

Author Response

The present study has described the synthesis and characterization of ternary Cu2O/CuS/ZnS nanocomposite using a two-step wet-chemical method for blue LED light-induced photocatalytic hydrogen production. The noble metal-free ternary nanocomposite has shown high photocatalytic activity in hydrogen evolution reaction. The manuscript is suggested for publication after minor modifications.

Response: Thanks for the pertinent and positive comments.

  1. The Cu-containing ternary catalyst shows high activity in photocatalytic hydrogen evolution. In order to demonstrate this, it would be very useful to compare the archived maximum value to that obtained on other noble metal-free ternary catalyst for example the recently obtained nickel phosphide anchored on anatase-hematite heterojunction (Int. Journal of Hydrogen Energy 47, (2022) 23593-23607). You can do it in the Introduction or in the Conclusion.

Response: Thanks for your reminder. We have added this reference in the introduction.

  1. The differentiation between Cu1+ and Cu2+ in XPS without analyzing the Auger transitions is very difficult.

Response: Thanks for your reminder. This suggestion will incorporate future research  in related XPS studies.

  1. The XPS technique is not mentioned in the 3.3 Characterization section.

Response: Thanks for your reminder. We have added the XPS technique (X-ray photoelectron spectroscopy (XPS, ULVAC-PHI PHI 5000 Versaprobe II system)) in the revised manuscript.

  1. Please indicate the BE reference in XPS (residual C?)

Response: Thanks for your reminder. The C 1s peak of residual C was used as the BE reference in XPS
